# Regular School Sport versus Dedicated Physical Activities for Body Posture—A Prospective Controlled Study Assessing the Sagittal Plane in 7–10-Year-Old Children

**DOI:** 10.3390/jcm11051255

**Published:** 2022-02-25

**Authors:** Mateusz Kozinoga, Łukasz Stoliński, Krzysztof Korbel, Katarzyna Politarczyk, Piotr Janusz, Tomasz Kotwicki

**Affiliations:** 1Department of Spine Disorders and Pediatric Orthopedics, University of Medical Sciences, 61-545 Poznan, Poland; k.politarczyk@gmail.com (K.P.); mdpjanusz@gmail.com (P.J.); kotwicki@ump.edu.pl (T.K.); 2Spine Disorders Center, 96-100 Skierniewice, Poland; stolinskilukasz@op.pl; 3Department of Physiotherapy, University of Medical Sciences, 61-545 Poznan, Poland; kkorbel@ump.edu.pl

**Keywords:** body posture, sports activity, corrective exercises, digital photography

## Abstract

Body posture develops during the growing period and can be documented using trunk photography. The study aims to evaluate the body posture in children aged 7–10 years undergoing a dedicated physical activities program versus regular school sport. A total of 400 children, randomly chosen from a cohort of 9300 participating in a local scoliosis screening program, were evaluated twice at a one-year interval. A total of 167 children were involved in regular school sport (control group), while 233 received both school sport and a dedicated physical activities program (intervention group). Standardized photographic habitual body posture examination was performed at enrollment (T0) and one-year after (T1). Sacral slope (SS), lumbar lordosis (LL), thoracic kyphosis (TK), chest inclination (CI), and head protraction (HP) were measured. At T0, the body posture parameters did not differ between groups. At T1 in the controls, all five parameters tended to deteriorate (insignificant): SS *p* = 0.758, LL *p* = 0.38, TK *p* = 0.328, CI *p* = 0.081, and HP *p* = 0.106. At T1 in the intervention group, the SS decreased (*p* = 0.001), the LL tended to decrease (*p* = 0.0602), and the TK, CI, and HP remained unaltered. At T1, the SS and LL parameter differed between groups statistically (*p* = 0.0002 and *p* = 0.0064, respectively) and clinically (2.52° and 2.58°, respectively). In 7–10-year-old children, participation in dedicated physical activities tends to improve their body posture compared to regular school sport.

## 1. Introduction

Harmonious arches characterize the shape of sagittal curvatures in a normal spine, which provides both mobility and stability [1]. Body posture develops throughout childhood and adolescence [2]. Increased sagittal spine curvatures (thoracic kyphosis or lumbar lordosis) combined with increased chest inclination and head protraction are observed in children with weak postural muscles [3] and in adults with myofascial pattern back pain [4]. The standardized photographic assessment of lateral trunk view can reliably quantify body posture and was validated for 7–10-year-old children [5]. The influence of physical activity on a child’s body posture has not been extensively studied so far. The study aims to assess the impact of dedicated physical activities versus regular school sports on a child’s body posture in a cohort of 7–10-year-old children.

## 2. Materials and Methods

### 2.1. Study Population

An observational, prospective controlled study was undertaken within the frame of a scoliosis screening program addressed to primary school children, inhabitants of a half million city. A total of 400 children, 200 girls and 200 boys, aged 7–10 years, were randomly selected from a base consisting of 9300 children, including all primary schools’ pupils of the city, previously negatively screened for idiopathic scoliosis or Scheuermann kyphosis. The technique of cluster randomization [6] was applied, assigning schools to districts first, then using a computer-generated sequence to select the order of districts randomly, and then the order of schools in individual districts. Inclusion criteria were age 7–10 years old, no structural deformation of the spine (i.e., idiopathic scoliosis or Scheuermann kyphosis), and availability of a full set of digital photographs. During school screening, the body posture assessment consisted of an inspection of spinal curvatures and position of the head. The assessments were performed by physiotherapists experienced in body posture examination. The examination was carried out according to a structured protocol. Physiotherapists received training before the procedure. Children were assigned into two groups: (1) the control group without any posture misalignment, who performed regular school sport and was considered natural history (N = 167), and (2) the intervention group of children presenting any posture misalignment, who received both school sport and dedicated physical activities (N = 233). The study was performed before the COVID-19 pandemic. The screening program was supported with a grant from the EEA Financial Mechanism. The study was approved by the local Institutional Review Board of the Poznan University of Medical Sciences, No. 283/16.

### 2.2. Intervention

The program for the prevention of postural misalignments (city of Poznan scoliosis screening program) comprised the participation of children in a system of organized dedicated physical activities. The group sessions, once per week, were organized and supervised by a physiotherapist. Several physiotherapists were engaged in the intervention, which could decrease the homogeneity of the intervention itself (study limitation). However, all of them received training and the dosage stayed fixed.

Children with postural misalignments (N = 233) entered in a weekly, one hour long, dedicated physical activities program, lasting ten months (September–June). Two major groups of activities were applied: either movement games and dance or swimming. All activities were supervised by a physiotherapist.

All children (N = 400) participated in regular 45 min long school sport activities (such as running, athletics, volleyball, basketball, handball, and football) four times per week, which is a standard frequency of school sport in primary schools in Poland.

### 2.3. Assessment

Each child underwent photographic registration of the body posture from the front (A), back (B), right lateral (C), and left lateral (D). The posture images were captured with a Canon^®^ (Tokyo, Japan) Power shot A590 IS digital camera (matrix CCD 1/2.5, 8.3 M pixels, focal length 35–140 mm). The camera was mounted on a tripod at the height of 90 cm and paralleled to the floor, 300 cm from the examined person. The photographic technique was standardized and consistent with the methodology published by Stolinski et al. [5].

The habitual standing posture of both groups was compared before (T0) and after one year (T1), at the end of the postural school program.

Five postural parameters were measured on digital photographs using a dedicated semiautomatic software SCODIAC v. 2.0 (Pavel Cerny, Prague, Czech Republic) [7] (Figure 1): (1) sacral slope angle (SS), (2) lumbar lordosis angle (LL), (3) thoracic kyphosis angle (TK), (4) chest inclination angle (CI), and (5) head protraction angle (HP). The increase of postural angular parameter was interpreted as a posture worsening, while decrease of postural angular parameter was interpreted as an improvement of body posture.

Clinically relevant differences were fixed as doubled SEM and revealed: SS 2.01°, LL 2.45°, TK 2.34°, CI 1.8°, and HP 0.89°.

### 2.4. Intra- and Inter-Rater Reliability

To determine the repeatability (intra-observer reliability), 40 digital photographs were evaluated three times in a one-week interval. Intraclass Correlation Coefficient (ICC), Confidence Interval (C.I.), Standard Error of Measurement (SEM), and Standard Deviation (SD) were calculated.

To determine reproducibility (inter-observer reliability), 30 digital photographs were evaluated by three observers three times each.

### 2.5. Statistical Analysis

GraphPad, MEDCALC, and Microsoft Excel were used. ICC values 0.81–1.00 were considered as excellent [8]. C.I. of 95% probability was taken. The normality of distribution was assessed using the Kolmogorov–Smirnov test. To assess the significant differences in average values, a t–Student test was used. The significance level was defined as *p* = 0.05.

## 3. Results

The repeatability (intra-observer reliability) for all five postural parameters was excellent (ICC 0.975 to 0.995), with SEM from 0.447° for HP to 1.225° for LL (Table 1).

The reproducibility (inter-observer reliability) was also excellent (ICC 0.885 to 0.975), with SEM from 0.888° for LL to 0.949° for HP (Appendix A, Table A1).

At study enrolment (T0), the habitual posture parameters SS, LL, TK, CI, and HP were not significantly different between both groups; the *p* level was 0.210, 0.366, 0.524, 0.757, and 0.611, respectively (Table 2).

The habitual posture assessment at one-year (T1 vs. T0) revealed differences in the control versus the intervention group. In the control group, small (from 0.17° to 0.96°), and statistically nonsignificant differences were found for all parameters (SS *p* = 0.758, LL *p* = 0.38, TK *p* = 0.328, CI *p* = 0.081, and HP *p* = 0.106, respectively). The values of all parameters tended to increase (Table 3).

In the intervention group, the SS significantly decreased (1.4°, *p* = 0.001), LL tended to decrease (1.59°, *p* = 0.062), and no significant change for the TK, CI, and HP was found (Table 4).

The habitual posture assessment at one-year (T1 intervention vs. T1 control) revealed statistically and clinically significant lower values of the SS and LL parameter (*p* = 0.0002, 2.52°; *p* = 0.0064, 2.58°, respectively) in the intervention group. The three other parameters did not differ between groups (Table 5).

## 4. Discussion

### 4.1. Body Posture Evaluation Technique

The photographic technique of the body posture evaluation presents an advantage to be non-invasive, objective, and easy to use (does not require the use of external skin markers). The terms of thoracic kyphosis, lumbar lordosis, or sacral slope denote the surface shape of the trunk but not the radiographic imaging. The methodology of photographic posture evaluation has been published [5,9,10,11], and the repeatability and reproducibility have been verified [5,9,12,13,14,15]. The previous study confirms excellent intra- and inter-reliability [5,9,10,11,12,13,14,15] and reflects the results by Stolinski et al. [5,15]. The ICC coefficient varied from 0.885 for the HP, up to 0.975 for the LL.

Archiving digital photography provides easy storage and reproduction of images, while respecting the patient’s anonymity. Canales et al. [16] used photographs of patients with mental disorders (depression) to document their postures. The technique allows for a combined photographic and clinical analysis of body posture, thus enhancing a child’s understanding of posture defect, which improves their cooperation during exercising.

### 4.2. Sample Selection

According to studies by McEvoy (U.S. population) and Shumway-Cook (Australian population) [13,17], the peri-pubertal growth spurt occurs at the age of 9–12 years (sooner for girls, later for boys). The study by Kułaga (Polish population) [18] in 2007–2012 on 22,000 children, aged 3–18 years, confirmed that the growth acceleration began at 9 years and 6 months for girls, and at 10 years and 3 months for boys. The Peak Height Velocity [19]—the most significant 12-month body height increase—equals 6.8 cm in girls and falls between 10 years and 8 months and 11 years and 8 months, while it equals 7.5 cm for boys and falls between 12 years and 6 months and 13 years and 6 months [18]. The rapid growth period corresponds to a period of deterioration of both structural spine deformities (idiopathic scoliosis) and non-structural ones (postural spine misalignments). Concerning the sample studied, the children presenting relevant trunk rotation during scoliometer examination had been previously identified and adequately managed. The remaining children were examined by physiotherapists and postural defects were noticed with inspection in about 40% of them. These children entered the dedicated program of physical activity in order to prevent further posture deterioration, improve motor skills, and enhance postural muscles force.

In this study, we examined the posture immediately before the critical period of the pubertal growth spurt. Comparing the values of the five postural photographic parameters between the two groups, no significant difference in the habitual posture at T0 was revealed.

### 4.3. Impact of Physical Activities on the Body Posture

In the control group (Table 3), no significant differences were noticed between the initial examination (T0) and the one-year examination (T1) for all parameters. All the values presented tendency to slightly increase. It can be supposed that in natural history (without the intervention of additional physical activities), postural parameters slightly deteriorate during the growing phase. Still, this change did not reach neither statistical nor clinically relevant values.

Among children from the intervention group (Table 4), examined in the habitual posture, between the initial examination (T0) and the one-year examination (T1), a significant (*p* = 0.001, Type II risk error 0.188) SS decrease of 1.40° was found, which should be interpreted as improved posture. The LL and HP parameter change was at the limit of significance (*p* = 0.062, Type II risk error 0.648; *p* = 0.058, Type II risk error 0.64, respectively). Other parameters did not change significantly, which can be interpreted as stabilization of postural photographic parameters in children from the intervention group.

Comparison of the habitual posture between both groups after one year (T1) has shown lower values of SS (*p* = 0.0002) and LL (*p* = 0.0064) in the intervention group, which signifies posture improvement. What is more, both differences were clinically significant (for SS: 2.52° (doubled SEM = 2.01°) and for LL: 2.58° (doubled SEM = 2.45°)) (Table 5).

The overall results indicate a positive effect of additional physical activities on a child’s body posture. The hypothetic cause could be a higher unconscious tension and activity of the abdominal muscles while standing.

### 4.4. Impact of Corrective Activities on Body Posture in the Literature

Dutkiewicz [20] examined the impact of 10-month corrective exercises among 370 children (190 study, 180 control) aged 6–17 years. The surface topography and Kasperczyk scoring method were used. Both groups at T0 did not differ. The highest reduction of posture defects was noted for the head position in the age group 6–9 years (from 23.6% to 13.9%) and 10–14 years (from 33.4% to 23.6% of prevalence). For children 6–9 years, the reduction of excessive thoracic kyphosis occurrence was significant (from 24.1% to 15.7%), as well as the reduction of excessive lumbar lordosis (from 23.8% to 18.8%).

The results of our study, similar to Dutkiewicz’s results, indicate an improvement in the thoracic kyphosis, however, we do not confirm the positive change in the head position and in lumbar lordosis angle.

Cosma et al. [21] reported on the influence of corrective exercises (static, dynamic, breathing, correcting, and hyper-correcting specific misalignments) on the body posture of 20 children aged 6–9 years. Body posture was assessed using the “PostureScreen Mobile” mobile application. The measurements were taken at the beginning of the study and after six months of physical exercises. The position of the head (protraction) and hip in the sagittal plane were assessed. A significant improvement of all measured postural parameters was noted. The study limitation, apart from a small study group, may be the fact that children were photographed in clothes, which makes it difficult to identify reference points on the body (such as sternum incision, acromions, waist, anterior superior iliac spines, greater trochanters, or external malleolus).

Torlaković et al. [22] published a study on the impact of physical exercises at the pool (swimming in combination with aquaerobics and corrective exercises) on the posture of 50 boys aged 5.2 ± 0.6 years in Sarajevo (Bosnia and Herzegovina). The posture was assessed by the Wolański visual method, assessing the position of the head, shoulders, shoulder blades, chest shape, spine, abdomen shape, knees, feet, and overall body posture. The program consisted of two weekly meetings of 60 min each for 16 weeks. Significant improvement in all parameters was noted, except for the chest shape and the head position. The limitation of the study is the use of a subjective method of assessing the children’s body posture.

### 4.5. Impact of Physical Activities on the General Health of the Child

The positive impact on body posture is not the only benefit coming from regular physical activities performed by children. The U.S. Department of Health and Human Services, in their Physical Activity Guidelines Advisory Committee Report, 2008 [23], stated that children and adolescents who maintain regular physical activity demonstrate a higher level of physical fitness (cardiorespiratory endurance and muscle strength), reduced body fat, favorable cardiovascular and metabolic disease risk profiles, enhanced bone health, and reduced psychological symptoms such as depression or anxiety. The positive impact of physical activities on bone mass was confirmed by Sundberg et al. [24], Welten et al. [25], and Bielemann et al. in the systematic review [26].

### 4.6. Practical Implications

We observe that regular school sport, even if extremely beneficial for physical development of children, may be insufficient to ensure a normal posture development due to the modern model of sedentary life. There is an important practical issue whether it could be enriched by out-of-school sport activities. Numerous initiatives of supporting children’s physical development are observed at the local, municipal, or regional level. Additionally, while strongly supporting children’s additional physical activities, we would like to emphasize the advantage of using objective tools (such as photographic posture assessment) for evaluating various programs raised throughout the country, especially while based on public funding.

## 5. Conclusions

(1) Participation in additional physical activities performed as a part of a dedicated program tended to improve body posture in 7–10-year-old children.

(2) Regular school sport activity may be insufficient as a compensation of the contemporary model of sedentary lifestyle in children and could be enriched by after-school sports activities.

## Figures and Tables

**Figure 1 jcm-11-01255-f001:**
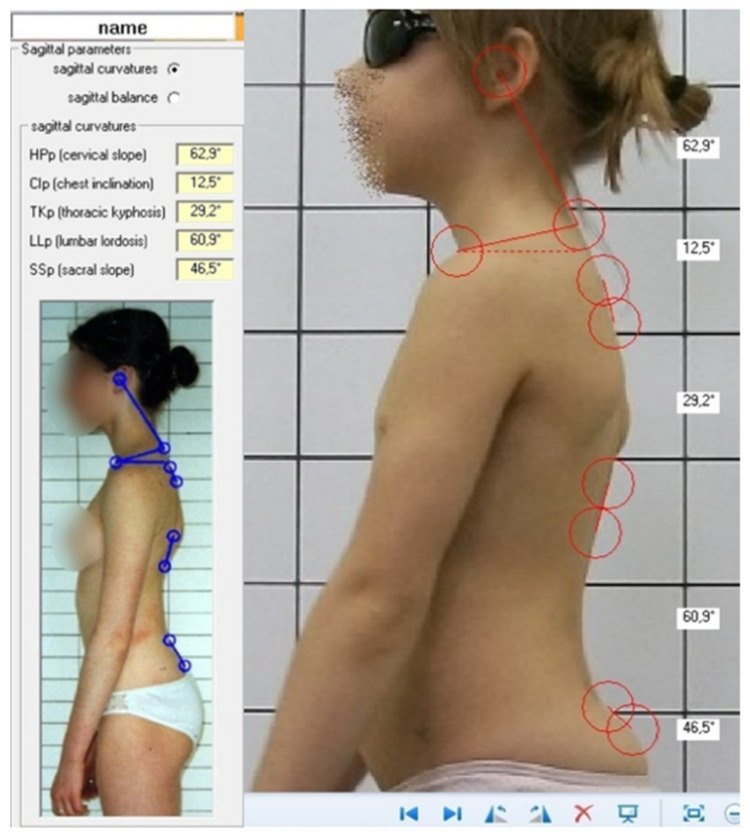
SCODIAC software v 2.0: top left—values of parameters; bottom left—instruction for use; right—measurement.

**Table 1 jcm-11-01255-t001:** Repeatability of photographic parameters measurements (Intra-observer).

Parameter	ICC	95% C.I.	SEM [°]	SD [°]
Sacral slope	0.975	0.884–0.986	1.007	6.4
Lumbar lordosis	0.981	0.968–0.989	1.225	8.88
Thoracic kyphosis	0.987	0.977–0.992	1.170	10.14
Chest inclination	0.981	0.968–0.989	0.901	6.55
Head protraction	0.995	0.991–0.997	0.447	6.32

ICC—Intraclass correlation coefficient, C.I.—Confidence Interval, SEM—Standard Error of Measurement, SD—standard deviation.

**Table 2 jcm-11-01255-t002:** Comparison of habitual posture between the control and the intervention group at T0 time.

	Posture	Control N = 167	Intervention N = 233	*p*
Parameter				
Sacral slope [°]		29.58 ± 7.23	28.63 ± 7.6	0.210
Lumbar lordosis [°]		46.94 ± 8.96	46.08 ± 9.78	0.366
Thoracic kyphosis [°]		41.63 ± 9.55	42.23 ± 9.11	0.523
Chest inclination [°]		15.9 ± 7	16.1 ± 5.64	0.757
Head protraction [°]		31.65 ± 5.16	31.96 ± 6.44	0.611

Mean value ± standard deviation is presented. N—group size, *p*—significance.

**Table 3 jcm-11-01255-t003:** Comparison of habitual posture between T0 and T1 in the control group (N = 167).

	Posture	H_0_	H_1_	*p*
Parameter				
Sacral slope [°]		29.58 ± 7.23	29.75 ± 6.79	0.758
Lumbar lordosis [°]		46.94 ± 8.96	47.50 ± 9.19	0.38
Thoracic kyphosis [°]		41.63 ± 9.55	42.38 ± 9.39	0.328
Chest inclination [°]		15.9 ± 7	16.86 ± 6.32	0.081
Head protraction [°]		31.65 ± 5.16	32.45 ± 6.08	0.106

Mean value ± standard deviation is presented. H_0_—habitual posture at T0, H_1_—habitual posture at T1, N—group size, *p*—significance.

**Table 4 jcm-11-01255-t004:** Comparison of habitual posture between T0 and T1 time in the intervention group (N = 233).

	Posture	H_0_	H_1_	*p*
Parameter				
Sacral slope [°]		28.63 ± 7.6	27.23 ± 6.68	**0.001**
Lumbar lordosis [°]		46.08 ± 9.78	44.49 ± 9.34	0.062
Thoracic kyphosis [°]		42.23 ± 9.11	42.79 ± 9.98	0.404
Chest inclination [°]		16.1 ± 5.64	16.37 ± 5.95	0.547
Head protraction [°]		31.96 ± 6.44	32.80 ± 6.07	0.058

Mean value ± standard deviation is presented. H_0_—habitual posture at T0, H_1_—habitual posture at T1, N—group size, *p*—significance.

**Table 5 jcm-11-01255-t005:** Comparison of habitual posture between the control and the intervention group at T1 time.

	Posture	Control N = 167	Intervention N = 233	*p*
Parameter				
Sacral slope [°]		29.747 ± 6.79	27.23 ± 6.68	**0.0002**
Lumbar lordosis [°]		47.50 ± 9.19	44.92 ± 9.34	**0.0064**
Thoracic kyphosis [°]		42.38 ± 9.38	42.79 ± 9.98	0.674
Chest inclination [°]		16.86 ± 6.32	16.37 ± 5.95	0.427
Head protraction [°]		32.45 ± 6.08	32.80 ± 6.07	0.57

Mean value ± standard deviation is presented. N—group size, *p*—significance.

## Data Availability

The data presented in this study are available on request from the corresponding author.

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
