# Peer review of "Regular School Sport versus Dedicated Physical Activities for Body Posture—A Prospective Controlled Study Assessing the Sagittal Plane in 7–10-Year-Old Children"

_jcm, 2022, doi:10.3390/jcm11051255_

Round 1
Reviewer 1 Report
WORK OF REMARKABLE SCIENTIFIC INTEREST, EXCELLENT RESEARCH WORK BOTH RELATED TO MATERIALS AND METHODS. EXCELLENT WORK OF STATISTICAL ANALYSIS. GOOD SCIENTIFIC RESEARCH, EXHIBITION SIMPLICITY AND THE NUMBER OF PARTICIPANTS IN THE RESEARCH. EXCELLENT BIBLIOGRAPHICAL REFERENCES. EXCELLENT MOTOR DESCRIPTIONS APPLIED TO THE STUDIO. JUST THE PURPOSE OF THE RESEARCH AND THE CONCLUSIONS.
Author Response
The authors sincerely thank the Reviewer for the time dedicated to reviewing this manuscript.

Reviewer 2 Report
This paper is interesting and provides additional data on a major topic in children's orthopedics. However, the article focused too much on the method and not enough on the outcomes. Further data analysis should be performed on the actual results for providing a precise meaning to the study. The construction of the paper lacks homogeneity. The discussion includes pieces of data that are mispositioned, and some of them should be in the results or the material and method section (see below).
MAJOR COMMENTS:
Amongst 200 children, more subgroups should be assessed, gender, height, BMI, for instance. Would you please perform subgroups analysis?
Would you please calculate the risk of Type 1 and Type 2 errors for all significant and nonsignificant data close to p=0.05?
What is the combined effect of sport and dedicated physical activities on the parameters measured? Please provide the global results as well ( compare TO to T1). What is the take-home message?
MINOR COMMENTS
Change the title by clarifying that it is an observational, prospective controlled study. Something like: "observational and prospective controlled study of 7-10-year-old children shows that ... "
If the standardized photographic assessment of lateral trunk view can reliably quantify body posture and was validated for 7–10-year-old children, why redo this in the method? If the results repeat what has been already found, please put this in supplemental data. If this is new, clarify it in the paper.
Would you please define what children with postural misalignments are?
Figure 1 is not appropriate for publication. Would you please describe the parameters in a clear picture?
"The body posture assessment consisted of a section of spinal curvatures performed during school screening by a physiotherapist."
Is it the same PT? Different one? How many? Did they receive training? Please clarify
Forty-five minutes long school sports activities four times per week; what does it include? aerobic exercise, I guess, please clarify
The discussion is confusing and mixed results and interpretation (e.g., Impact of physical activities on the body posture). The intervention paragraph should be in the material and method. Would you please reorganize the discussion to make this shorter and provide a clear take-home message?
The conclusion is that dedicated physical activities tend to improve body posture based on SS and LL; what does "improve" mean? To which extent?
Some publications related to children bone growth should be cited in the introduction/discussion, such as:
Sundberg, M., Gärdsell, P., Johnell, O. et al. Physical Activity Increases Bone Size in Prepubertal Boys and Bone Mass in Prepubertal Girls: A Combined Cross-Sectional and 3-Year Longitudinal Study . Calcif Tissue Int 71, 406–415 (2002). https://doi.org/10.1007/s00223-001-1105-z
Welten DC, Kemper HC, Post GB, Van Mechelen W, Twisk J, Lips P, Teule GJ. Weight-bearing activity during youth is a more important factor for peak bone mass than calcium intake. J Bone Miner Res. 1994 Jul;9(7):1089-96. doi: 10.1002/jbmr.5650090717. PMID: 7942156.
Bielemann RM, Martinez-Mesa J, Gigante DP. Physical activity during life course and bone mass: a systematic review of methods and findings from cohort studies with young adults. BMC Musculoskelet Disord. 2013 Mar 4;14:77. doi: 10.1186/1471-2474-14-77. PMID: 23497066; PMCID: PMC3599107.
Berteau, JP. Biomechanics of growing bone, a support to pediatric physiotherapy Kinésithérapie, la revue November 2013 13(143):16-21 doi.org/10.1016/j.kine.2013.06.014
Reviewer 3 Report
The authors have done a good job monitoring and following Children regarding body posture, I have some questions and comments I hope can be clarified.
1) It is not stated regarding the Clinical experience of the screening physiotherapist. Additionally, do the authors Believe it would be beneficial using more than one person doing the screening?
2) In the comparisons of angles there are in many cases difference of only 1-2 degrees; do the authors take into account margins of error? It seems like this can influence the results.
3) Were the included Children physical Active in their spare time? Did they do other types of sports and was this accounted for in the analysis? It is not clear also whether all Children completed follow-ups, it would be of interest to know the drop-out rate from baseline to follow-up.
4) I do agree with the authors conclusions regarding benefits of physical activity; however posture is not the main benefit. Mental, skeletal and physical benefits from physical activity are well-documented and I suggest the authors add these benefits in the discussion. It should also be pointed out that posture is individual and a few angles deviating from what is considered normal is not suggestive of pathology or disturbance in the growing Childs body.
Round 2
Reviewer 2 Report
Please integrate The risk of making Type 2 errors in your discussion and also to moderate your results.
Where are the calculation for table Comparison of habitual posture between T0 and T1 time in the intervention group, N = 233. This is the most important in terms of Type 1 and Type 2 errors.
In addition, when there is statiscal difference please provide the power of the results. A power has to be high enough to show the benefit of the intervention.
Table 2. CI parameter, p=0.081. Risk of making a Type II error 0.585 (58%).
Table 3. SS parameter, p=0.001. Risk of making a Type II error 0.188 (18.8%)
LL parameter, p=0.062. Risk of making a Type II error 0.648 (64.8%)
HP parameter, p=0.058. Risk of making a Type II error 0.064 (64%)
Author Response
The authors sincerely thank the Reviewer for the time dedicated to reviewing this manuscript and valuable comments. The answers are presented below in red.
-Please integrate The risk of making Type 2 errors in your discussion and also to moderate your results.
We integrate the risk of Type 2 error in the discussion. Please see line 205.
-Where are the calculation for table Comparison of habitual posture between T0 and T1 time in the intervention group, N = 233. This is the most important in terms of Type 1 and Type 2 errors.
We made a mistake in the first answer putting the risk of Type 2 error as a calculation for Table 3 (for the control group) instead of Table 4 (for the intervention group). Apologies. The calculation for the risk of Type 1 and 2 errors for the study group are as follow:
We set the Type 1 risk error as 0.05 (5%)
Type 2 risk error:
SS parameter, p=0.001. Risk of making a Type 2 error 0.188 (18.8%). Power of the result 0.0812.
LL parameter, p=0.062. Risk of making a Type 2 error 0.648 (64.8%). Power of the result 0.325.
HP parameter, p=0.058. Risk of making a Type 2 error 0.64 (64%). Power of the result 0.360.
-In addition, when there is a statistical difference please provide the power of the results. A power has to be high enough to show the benefit of the intervention.
Please see above.
Submission Date 2021-09-17. Date of this review 16th of February 2022
Reviewer 3 Report
Although having some methodological issues previously raised, the authors have done a good job revising the manuscript.
Author Response

(The authors gave the same response as above.)
